# “Do Not Mix Apples with Oranges” to Avoid Misinterpretation of Placebo Effects in Manual Therapy: The Risk Is Resulting in a Fruit Basket. Comment on Molina-Àlvarez et al. Manual Therapy Effect in Placebo-Controlled Trials: A Systematic Review and Meta-Analysis. *Int. J. Environ. Res. Public Health* 2022, *19*, 14021

**DOI:** 10.3390/ijerph20156444

**Published:** 2023-07-26

**Authors:** Giacomo Rossettini, Leonardo Pellicciari, Andrea Turolla

**Affiliations:** 1School of Physiotherapy, University of Verona, 37134 Verona, Italy; 2IRCCS Istituto delle Scienze Neurologiche di Bologna, 40139 Bologna, Italy; 3Department of Biomedical and Neuromotor Sciences (DIBINEM), Alma Mater Studiorum University of Bologna, 40126 Bologna, Italy; 4Unit of Occupational Medicine, IRCCS Azienda Ospedaliero-Universitaria di Bologna, 40138 Bologna, Italy

We read with interest the systematic review with the meta-analysis by Miguel Molina-Álvarez et al. [1], recently published in the International Journal of Environmental Research and Public Health. We want to thank the authors for analysing the role of placebo effects and the design of placebo control groups in manual therapy, thus allowing us to contribute to the scientific debate on such a relevant topic for clinicians. Unfortunately, despite the authors’ best willingness, several methodological issues in the published systematic review seem to undermine the research’s soundness by weakening the credibility of results and the certainty of conclusions. Accordingly, we would provide a critical analysis suggesting point-by-point potential solutions to the major criticisms.

Firstly, the authors include patients “suffering from pain... regardless of the characteristics of the participants”, considering both musculoskeletal disorders (e.g., low back pain) and non-musculoskeletal (e.g., primary dysmenorrhoea) [2] conditions of medical relevance. The heterogeneity of clinical conditions presented, referable to different etiologies (e.g., acute and chronic pain) and neurophysiopathological mechanisms (e.g., nociceptive, nociplastic, and neuropathic), potentially affects the analysis of placebo effects in the sham group, introducing critical biases [3]. As a possible solution, the authors could have only considered musculoskeletal pain conditions of interest for manual therapy (e.g., low back pain), following the practice of recent evidence [4], or tried to explore by population (sub-groups analysis) any different effect.

In this regard, a second issue is related to subgroup analyses conducted, pooling studies according to the same treatment techniques (e.g., massage, neural mobilization, manipulations) but not grouping studies investigating the same condition. Indeed, in the same meta-analysis, studies on patients with chronic low back pain and carpal tunnel syndrome were pooled together. However, the role of psychosocial factors is more critical in patients with chronic low back pain [5] than in patients with carpal tunnel syndrome [6]. One of the essential assumptions of the meta-analysis was to group studies homogeneously [7]; thus, “apples” (e.g., subjects with chronic low back pain) should have been analyzed separately from “oranges” (e.g., subjects with carpal tunnel). In this way, it would have been possible to have homogeneous results, more valuable to clinicians for precise indications of the role of the placebo effect in specific conditions. Conversely, the inclusion of apples and oranges together increased the risk of heterogeneous results that were difficult to interpret in the absence of exploration based on clinical needs. As a solution, we suggest that the authors perform subgroup analyses, grouping more homogeneous studies in terms of the population examined, as advised by other authors [8].

Thirdly, the authors report in the literature search examples of “combinations of medical terms (MeSH) and keywords” used missing to report explicit syntax used for each database. The reporting of “full search strategies for all databases, registers, and websites, including any filters and limits used” is required by the PRISMA guidelines of 2020 [9], to which the authors refer in their systematic review. Since proper search databases for the literature is a milestone in a systematic review, a poor search strategy may lead to low-quality evidence undermining its transparency and reproducibility [9]. As a solution, the authors could have developed search strategies following the Peer Review of Electronic Search Strategies (PRESS) guidelines [10] that provide complete direction for structuring strings within the systematic reviews.

Fourthly, the authors state that they only include “parallel RTCs”. However, their result reveals other designs, such as a controlled trial [11]. This heterogeneity of included studies (e.g., with or without randomisation) to answer a specific question may lead to bias resulting in uncertainty on the interpretation of the direction and magnitude of treatment effect [12]. As a solution, the authors should have included only randomised controlled trials as stated in their PROSPERO protocol (ID: CRD42020157468). Alternatively, they could have declared deviations from it, thereby improving the transparency of their reporting or including only RCTs in the meta-analyses.

Today, manual therapy is experiencing a constitutive refoundation, including the management of placebo and nocebo effects as a new core competence [13]. Within this transition, there is an emerging scientific debate aimed at identifying proper placebo comparators [14,15], with large work still needing to be done to solve critical issues. Along time, RCTs are increasing in the field, but the methodology of conducting and reporting is improving accordingly [16]. Furthermore, biases and limitations, such as small sample size, inadequate study length, short-term follow-up, and inadequate controls (placebo or sham), undermine the internal and external validity of RCTs in manual therapy [17]. In addition, control groups have several areas to be improved, such as maintaining conditions comparable to experimental interventions or ensuring the inertness of the sham technique [18,19]. Furthermore, a non-manual comparator (e.g., modalities-detuned ultrasound or laser) is often applied in RCTs instead of using proper manual comparators (e.g., sham touch, shame joint mobilization), which lack the characteristics of indistinguishability necessary to define it as ‘placebo’ [20].

Taken together, these findings may discourage the search for a placebo comparator, considering it more of a chimaera than a need. However, the placebo control group is essential in non-pharmacological RCTs (e.g., in manual therapy) to inform patients, clinicians, and policy-makers about the efficacy and risks of interventions [21]. Therefore, we call to action researchers in manual therapy to improve the reporting of their trials by providing more details on placebo control (e.g., procedure, dose), controlling for non-specific elements (e.g., expectation, rituals), and ensuring the indistinguishability of the technique (e.g., comparing real manual techniques with sham techniques and not with other modalities). For example, the extension of the Template for Intervention Description and Replication for Placebo-controlled trials (TIDieR-Placebo) [22] is a starting point to improve reporting. However, more suggestions for non-pharmacological RCTs (e.g., manual therapy) are needed. As a scientific community, moving in this direction will open the opportunity to strengthen scientific evidence, and shift the debate towards solutions resulting in better patients care and growth for all clinicians interested in manual therapy.

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
