# Peer review of "“Do Not Mix Apples with Oranges” to Avoid Misinterpretation of Placebo Effects in Manual Therapy: The Risk Is Resulting in a Fruit Basket. Comment on Molina-Àlvarez et al. Manual Therapy Effect in Placebo-Controlled Trials: A Systematic Review and Meta-Analysis. Int. J. Environ. Res. Public Health 2022, 19, 14021"

_ijerph, 2023, doi:10.3390/ijerph20156444_

Round 1
Reviewer 1 Report
Article needs to be improved.

Author Response
Dear Editor-in-Chief and Reviewers,
We would like to thank for your interest in our manuscript submitted to International Journal of Environmental Research and Public Health (ijerph-2198884: "Do not mix apples with oranges" to avoid misinterpretation of placebo effects in manual therapy: the risk is resulting in a fruit basket.).
We read the comment of the Reviewers; however, to date, we did not made any changes to our commentary for the following reasons:
- a) Reviewer#1 raised critical issues that we do not agree with,
- b) Reviewer#2 did not report any critical issues.
However, we have made clarifications for each issue raised.
Please find below our point-by-point responses to the peer reviewer's comments. Thank you again for allowing us to revise the manuscript in-depth for your full consideration.
With best wishes,
Leonardo Pellicciari
On behalf of authors
Reviewer#1’s comments:
Thank you for the opportunity to review this article. The authors have taken up an important and interesting topic.
Reply: Dear Reviewer#1, Thanks a lot for your interest in our manuscript (ID ijerph-2198884: “Do not mix apples with oranges” to avoid misinterpretation of placebo effects in manual therapy: the risk is resulting in a fruit basket.). Although you provided some interesting feedback, we decided not to make changes in our letter to the editor. Please find below our point-by-point responses to the peer reviewer's comments. Thank you again for allowing us to revise the manuscript in-depth for your full consideration.
R1.1. The authors of the publication point out the inadequacies of the analyzed article, and they should also state what and how the analyzed article should be improved, so that the analyzed article is well described/presented. Why the authors focused only on analyzing 1 article?
Reply: Thanks for your comment. Our 'commentary' is a letter to the editor analyzing the limitations of the review by Molina-Alvarez et al. Accordingly, we have only analyzed this article.
R1.2. Line 17 - The authors are only referring to the article Miguel Molina-Alvarez? There are many publications about specific and non-specific LBP.
Reply: Thanks for the request. As it is a letter to the editor, we have only focused on one paper (Molina-Alvarez et al.) and analyzed its various criticalities.
R1.3. Line 27 - The authors make an allegation that they should describe in more detail.
Reply: We have quoted in inverted commas "" a portion of text directly taken from the article by Molina-Alvarez et al., which is the object of analysis in our letter to the editor.
R1.4. Line 54-63 - The authors undermine - fine, but they should give where and how to look, give a link, a journal and etc.
Reply: The points raised always concern the article by Molina-Alvarez et al., that is the object of analysis of our letter to the editor.
R1.5. Limitations - they should be added, are the authors sure they didn't have them when writing this comment?
Reply: Thank you for the suggestion. As it is a letter to the editor, it does not need a specific section named "limitation".
R1.6. References - Why did the authors refer to publications from before 2013?
Reply: The use of references in this letter is intended to support the critical issues we discussed. Therefore, we have not set time parameters to select references relevant to our discussion, as it is a letter to the editor.
R1.7. Maybe the authors of this article should write tips for other authors on how to prepare good articles: metanalysis, review and etc.
Reply: Thank you for the request. As you will see, we proposed solutions that could have been adopted to improve the draft for each point of criticism raised. You will find our suggestions at the end of each paragraph.
R1.8. Preparing a good metanalysis, contrary to appearances, is not easy.
Reply: We recognize that preparing and conducting a meta-analysis is a challenging process. Accordingly, we suggested to the authors the importance of using and following the reporting guidelines (PRISMA statement) shared by the international scientific community.

Reviewer 2 Report
Although I did not review the original article, the methodology in general is very much in line with what we usually do in our environment (Spain). However, I agree with you in your general idea that the analysis should have been segmented by groups, that the pain characteristics of the patients influence the outcome and that it would have been more appropriate to include only musculoskeletal pain conditions. I therefore consider your comment to be pertinent.
Author Response
Dear Editor-in-Chief and Reviewers,
We would like to thank for your interest in our manuscript submitted to International Journal of Environmental Research and Public Health (ijerph-2198884: "Do not mix apples with oranges" to avoid misinterpretation of placebo effects in manual therapy: the risk is resulting in a fruit basket.).
We read the comment of the Reviewers; however, to date, we did not made any changes to our commentary for the following reasons:
- a) Reviewer#1 raised critical issues that we do not agree with,
- b) Reviewer#2 did not report any critical issues.
However, we have made clarifications for each issue raised.
Please find below our point-by-point responses to the peer reviewer's comments. Thank you again for allowing us to revise the manuscript in-depth for your full consideration.
With best wishes,
Leonardo Pellicciari
On behalf of authors
Reviewer#2’s comments:
R2.1. Although I did not review the original article, the methodology in general is very much in line with what we usually do in our environment (Spain). However, I agree with you in your general idea that the analysis should have been segmented by groups, that the pain characteristics of the patients influence the outcome and that it would have been more appropriate to include only musculoskeletal pain conditions. I therefore consider your comment to be pertinent.
Reply: Dear Reviewer#2, Thanks a lot for your interest in our manuscript (ID ijerph-2198884: “Do not mix apples with oranges” to avoid misinterpretation of placebo effects in manual therapy: the risk is resulting in a fruit basket.). We thank you for your positive comments and are confident that the paper will be published soon. Thank you again for allowing us to revise the manuscript in-depth for your full consideration.
